# Enhancing our understanding of short-term rental activity: A daily scrape-based approach for Airbnb listings

**Yang Wang**[iD]*, **Mark Livingston**[iD], **David P. McArthur**[iD], **Nick Bailey**[iD]

Urban Big Data Centre, University of Glasgow, Glasgow, Scotland, United Kingdom

* yang.wang@glasgow.ac.uk

**Data Availability Statement:** We wholeheartedly appreciate the journal's dedication to openness in research methodologies and data sharing, as we believe this approach greatly benefits the broader

## Abstract

The growth of the online short-term rental market, facilitated by platforms such as Airbnb, has added to pressure on cities' housing supply. Without detailed data on activity levels, it is difficult to design and evaluate appropriate policy interventions. Up until now, the data sources and methods used to derive activity measures have not provided the detail and rigour needed to robustly carry out these tasks. This paper demonstrates an approach based on daily scrapes of the calendars of Airbnb listings. We provide a systematic interpretation of types of calendar activity derived from these scrapes and define a set of indicators of listing activity levels. We exploit a unique period in short-term rental markets during the UK's first COVID-19 lockdown to demonstrate the value of this approach.

## Introduction

Short-term rental (STR) platforms have been disruptive not only to the hospitality industry [1–3] but also to the housing markets and to the neighbourhoods most directly impacted [4–8]. As countries emerge from the COVID-19 pandemic, these disruptions may increase as cities seek to reap the economic benefits of tourism which STR may encourage. Many city authorities have started revising regulations and policies, trying to balance the economic contributions of the STR sector and the associated negative externalities. Now, more than ever, we need accurate data on the STR market to understand these impacts and design effective regulations to mitigate them.

Unfortunately, platforms such as Airbnb appear unwilling to share data on their listings and activity levels. Indeed, they often take steps to obscure activity [9]. As a result, researchers and local governments have to rely on third-party data providers to try to understand what is happening. For Airbnb, still the dominant platform [10], there are two main distributors of data. AirDNA provides a more comprehensive dataset for which they charge users. The main issue for researchers is that the processes of scraping and development are not shared so it is not possible to know the veracity of the data and metrics provided. InsideAirbnb, on the other hand, provides free access to the data it scrapes from the Airbnb website. The code they use to scrape data and produce metrics is public giving transparency but the data is relatively patchy both temporally and spatially, reflecting their much more constrained resources. The

research community. In our paper, we propose an open methodology that critically examines short-term rental market data obtained by employing web scraping techniques on the Airbnb platform. Our primary objective is to demonstrate the value of this approach for researchers who are otherwise solely reliant on proprietary data or open data which are much more limited in scale (see paper for details). Our scraping exercise accesses the openly-available Airbnb listings data under the provisions of UK copyright law. In the UK, the text and data mining exemption to copyright law permits the collection, storage and analysis of data which researchers have a legal right to view (in this case, Airbnb's public property listings) where the purpose is non-commercial academic research. However, the law does not permit the sharing or distribution of the raw data to others. For work in our field which uses this approach to data collection, it is therefore not possible for researchers to provide direct access to the raw data they have used. For a detailed discussion of the legal issues, please see: Burrow, S. (2021) The Law of Data Scraping: A review of UK law on text and data mining. CREATe Working Paper 2021/2 (https://doi.org/10.5281/zenodo.4635759). Our team at Urban Big Data Centre has provided details of the methods and code used to collect the data and these are available to others so they can create their own data collections (details here: https://github.com/urbanbigdatacentre/ubdc-airbnb/tree/master/README). In light of our commitment to transparency, we had made two additional steps to ensure that our paper meets the journal's requirement of making minimal data fully available: (a) We make the code used to process the data and generate our analyses accessible through GitHub [https://github.com/YangWang-Glasgow/Airbnb-Processing-Daily-Booking-Calendars]. This approach would provide complete transparency regarding our methods, allowing other researchers to scrutinize, repeat and build upon our work. It cannot support direct replication however since we cannot legally share the dataset used in the paper. (b) Additionally, we share the aggregated data used in the creation of our figures as part of the Supporting Information files. This step would facilitate a understanding of our findings and enable other researchers to utilize this aggregated data for further analyses or validation. We hope these proposed measures align with the journal's data availability requirements and they would be considered sufficient to ensure the minimal data necessary for replication and validation are fully accessible to the research community. We are unable to go further while remaining within the UK law.

calculation of key indicators, such as occupancy, therefore necessarily relies on some significant assumptions [11].

A more transparent, comprehensive and open approach is needed to address these data gaps. Ideally, it should: let other researchers scrutinize and reproduce the process end-to-end; provide a means for assessing the existing data from AirDNA and InsideAirbnb; and derive more fine-grained data on daily market activities. The Urban Big Data Centre (UBDC) provided the basis for this work, developing an openly available (https://github.com/urbanbigdatacentre/ubdc-airbnb/tree/master/README) framework for scraping listings on a daily basis from platforms. UK law allows the scraping of websites for research purposes although there is debate about whether the resulting data can be legally shared with others [12]. Using this framework, this paper aims to show how it can be employed to construct a database from a daily scraping of the Airbnb website which supports a much more fine-grained analysis of listing activity. This paper focuses on the methods used to process the scraped data and to interpret the information to derive more accurate estimates of key market indicators than that are available from existing sources.

To demonstrate the value of this approach, we apply it to the city of Edinburgh during an exceptional period in STR market activity: the first five months of the UK's COVID-19 lockdown from March 2020 onwards. These months saw Airbnb introduce a series of policy interventions, the 'Extenuating Circumstances Policies' (ECPs), in response to the sudden restrictions on mobility which prevented people from using bookings they had made. The ECPs permitted people to cancel bookings without penalty, regardless of booking conditions. This period is particularly useful as a test of our method. The rapid succession of policy changes posed challenges not only to distinguish and evaluate the impact of the policies implemented at different stages, but also to define and explain noise and uncertainties. We show that the proposed method provides a better picture of market activities compared with scrapes with larger intervals. Daily scraping plays a crucial role in ensuring the quality of the data, especially as it is being examined for the first time in this context. We believe this provides confidence that the methods developed can be used to underpin future research in this important field and provide a firmer foundation for policy.

## Background

The rapid growth of online home-sharing platforms has made STR increasingly popular worldwide. By bringing together hosts, who have a spare bedroom or property, and guests, who seek short stays, platforms make the short-term rental process efficient on a global scale. As originally promoted, at least, platforms enable guests to find 'authentic' but affordable places to stay [13] while hosts earn extra income [14], with platform providers taking a share for their services.

However, people outside this 'triad' of host-renter-platform may have to bear the negative externalities of this activity. A large volume of tourists staying in residential buildings or areas introduces extra noise, waste and traffic congestion, especially during periods of peak demand [15, 16]. Misuse of properties can cause additional issues with some reportedly used as party flats, even during the pandemic, breaching social distancing laws [17]. As properties are removed from the housing system, long-term residents worry about losing their sense of the community, as well as their quality of life and well-being [18]. While some research has found positive impacts of Airbnb activities with, for example, increased tourism [19], better urban amenities [20] or greater employment in restaurants [21], negative impacts may include the loss of amenities valued more by long-term residents such as local shops, post offices, banks or libraries [22]. As Airbnb restricts access to their data, the lack of evidence on host activity levels

**Funding:** This work has funding support from ESRC (ESRC-funded Urban Big Dat Centre (UBDC) [ES/L011921/1 and ES/S007105/1]). The funders had no role in study design, data collection and analysis, decision to publish, or preparation of the manuscript.

**Competing interests:** The authors have declared that no competing interests exist.

leaves the local authorities with a challenge in collecting any occupancy taxes which might otherwise have offset impacts on local public services such as waste disposal [23].

## Housing impacts of short-term rentals (STR)

The most significant challenge faced by many local authorities is the impact of short-term rental on a city's housing system. Such impacts are multifaceted. Firstly, STR listings may drive up long-term rental prices as renters compete for the reduced supply in that market. Barron et al. [4] found a 1% increase in Airbnb listings leads to a 0.018% increase in rents in the US. This ratio is even higher in areas with a low owner-occupancy rate. Wachsmuth et al. [6] found Airbnb has increased the median long-term rent in New York City by 1.4%, translating into $380 rent increase, over 2014–2017. Similarly, Horn & Merante [5] found one standard deviation increase in Airbnb listings is associated with an increase in asking rents of 0.4% in Boston. In London, it was found that an 8% increase in unit rental price per bedroom per week translates to £90 price increase per year [24] if the density of misuse listings on Airbnb were doubled.

Secondly, STR may inflate house prices. Barron et al. [4] report a 1% increase in Airbnb listings leads to a 0.026% increase in house prices for the zipcode with the median owner-occupancy rate in the US. Employing hedonic models, Sheppard & Udell [25] estimate a doubling of Airbnb listings is associated with increases of 6% to 11% in house values in a 300-meter zone in New York City while Zou [8] reports a 0.78% increase in property prices for each additional Airbnb listing within the 200-foot buffer in Washington, DC.

Last but not least, the emergence of a large number of commercial operators who manage multiple STR listings and offer full-time rental services may reduce a city's housing supply. Local studies in many cities support this speculation. In the US, O'Neill & Ouyang [26] conducted a multi-city analysis and found that commercial operators are key players in the Airbnb market, contributing 40% of total revenue. San Francisco [27] found that: 57% of listings were rented entirely without host presence; 64% of listings in Los Angeles were not occupied by owners [28]; commercial hosts control half of the total listings in Boston [5]; and 12% hosts in NYC of this category earn over 28% revenue. Similarly, in the UK, the Greater London Authority report [29] shows that around 16% (5260) of commercial hosts manage around 45% (21,440) listings. Among this group, just 280 hosts each with more than ten properties accounted for 15% or 7440 of the active STRs. Scottish Government's recent analysis also showed that a very small proportion of hosts (0.3%) manage a large proportion of total listings (13%), with portfolios ranging from 16 to over 100 properties [30].

The presence of a relatively small number of commercial hosts who dominate activity indicates a capital shift from the long-term rental (LTR) market to STR. The Airbnb-induced rent gap [6, 31] offers landlords the potential for higher financial returns from STR while maintaining the possibility to buy and sell quickly [32]. The uneven geographic distribution of Airbnb listings [15, 33], often clustered into the city centre and tourist hot spots, creates localised housing affordability issues that worsen problems of displacement and spatial inequality [28].

## Researching short-term rentals (STR): Data sources and methods

The emergence of STR platforms is just one example of the ways in which the digital revolution is impacting on society and driving the emergence of new forms of data. These 'digital footprints' provide a range of new opportunities for researchers to study cities. They encompass user-generated content and data from sensors as well as that produced in the digital systems of businesses such as STR platforms and public services [34]. While there are many potential advantages with these sources compared with traditional quantitative data (usually

the household survey or Census), they also come with important limitations. Many of these stem from the fact that data production and ownership have shifted from the public or academic sectors to the private [35]. This creates challenges in accessing data which is now viewed as a commercial asset. It also raises issues with assessing the quality of data products–and the research built on them–since methods are frequently obscured, ostensibly for reasons of commercial confidentiality but perhaps also to hide problems with data quality [36].

Airbnb is a good example of a private company that restricts access to its data [9] although here the motivation may be as much about its concern to limit regulatory intervention as its desire to keep information on market activity hidden from rivals. There have been several legal battles between US city governments and Airbnb over access to transaction records for regulation purposes. Hoffman and Heisler [9] describe the cases in New York City, Boston and San Francisco as 'data wars' during which Airbnb refused to disclose data, released only selected data and obstructed independent analysis [37]. The public-facing platform has also been redesigned at times in ways which make it more difficult to monitor activity levels [38].

In this context, there are three main routes to data on Airbnb activity: data provided by Airbnb itself; proprietorial data products, notably those produced by AirDNA; or third-party web scraping which underpins the open data from InsideAirbnb.

**Airbnb transaction data.** After becoming a publicly-listed company at the end of 2020, Airbnb is expected to be more open about its data [39]. They have set up the Airbnb City Portal to provide local authorities with more access to income of listings and hosts and assess if they are complying with the local housing or tax regulations. Airbnb report that over 300 cities and tourism organisation have accessed the portal [40]. The data, however, is not open to wider research use at this time. It remains to be seen how much detail will be provided.

**Proprietorial data.** A popular secondary dataset used in previous research comes from AirDNA, a consultancy company providing professional investment advice to potential/existing Airbnb hosts using their scraped online platform data. Researchers are charged to access the processed data but these provide a detailed picture of daily market activities. In addition to the information about each listing, including booking calendar, reviews and geographical neighbourhood, one of the advantages of AirDNA data is their estimate of the listing's daily occupancy rate which is key to estimating likely revenues and hence potential returns for investors. As a result, AirDNA data has become a major data source to facilitate revenue-based analysis including work on housing market impacts [31, 41, 42]. The identification of 'vulnerable neighbourhoods' most at risk of an expansion in Airbnb activity relies on comparing the revenue potentially earned from listing in the STR market with those from the traditional LTR.

The issue with the estimates from AirDNA is the lack of transparency in the methods which raises concerns about quality. AirDNA state that they base their methods on occupancy levels which could be observed directly up until 2014 when Airbnb changed its website to make it impossible to distinguish the days a listing is 'booked' from those when it is 'unavailable' for other reasons. AirDNA built machine learning models to predict occupancy-based data from this earlier period. However, their methods are not openly available for scrutiny although they offer academic products for academics and the training data which underpin their methods is increasingly out-of-date.

**Third-party scraping.** Platform activities rely on publicly-accessible information on listing availability and prices, providing an opportunity for 'scraping' or collecting data using automated processes. Third-party web-scraped data have been made available using this approach. Most notably, InsideAirbnb, hosted by Cox, become a widely used Airbnb data source helping many city regulators (e.g. New York, San Francisco, and Scotland) and facilitating much research e.g. Hoffman & Heisler [9] and Zou [8].

InsideAirbnb data currently covers over 80 cities worldwide. It comprises comprehensive information about listings, their locations and neighbourhoods, structural characteristics and amenities, booking calendars, policies and requirements, basic information about hosts and guests, and guests' comments. However, the released scrapes are snapshots of the market on particular days, not continuous streams of listing traces. Calendar updates are made available monthly (or even less frequently in some cities). This makes occupancy estimation more difficult because bookings and cancellations may occur between scrapes as people make last-minute changes. Researchers, therefore, have to make many assumptions to estimate occupancy.

Due to this limitation, researchers rarely used InsideAirbnb to estimate detailed market activities from the booking calendars. As far as we are aware, the most recent research to use InsideAirbnb booking histories to analyse occupancy rates is Boros, Dudás, & Kovalcsik [43]. It compares listings' calendar updates between consecutive monthly data releases to learn about the growth and loss of bookings during the pandemic. As we show below in our analysis, however, the lack of detailed calendar activities is likely to lead to an underestimation of the volumes of occupancy change. There is still a gap in a systematic interpretation of the meaning and limitations of such estimations from the calendar.

## Short-term rental (STR) during the pandemic

The Covid-19 pandemic brought an unprecedented shock to STR. Like other parts of the tourism industry, these platforms were hit hard by near-global travel restrictions. In recognition of the unique circumstances, platforms responded by offering customers the chance to cancel bookings made prior to the pandemic without penalty, regardless of prior contractual terms [44]. Airbnb referred to these as its 'Extenuating Circumstances Policies' (ECPs). This provoked some anger among hosts since they were ultimately the ones who suffered resulting losses [45] and it is possible this will contribute to the reshaping of the STR market [46].

As Airbnb were doubtless aware of the likely impact on owners, the ECPs were introduced in phases, each extending cancellation rights for a limited period and reflecting restrictions in place in different localities around the world; detailed restrictions in Scotland and the ECP phases for the UK are described in S1 Appendix. In the first phase, the ECP allowed hosts and guests to cancel bookings for the period 14 March to 14 April 2020 without charge or penalty where the booking was made on or before 14 March 2020. As the pandemic deepened in the UK, the ECP was updated several times through to the end of October 2020. This paper focuses on the five earliest ECPs, Policies 1–5. Fig 1 shows a detailed timeline with spikes indicating the date a new Policy was announced and bars of the same colour indicating the period of bookings covered as a result. Each time the market reacted differently with varying temporal and financial trends. This is the first reason that this period is so valuable for developing and testing a method to monitor market activity.

The second reason to choose this period is because of the overlapping coverage of the Policies. Every time a new extension was announced, it added additional coverage of bookings

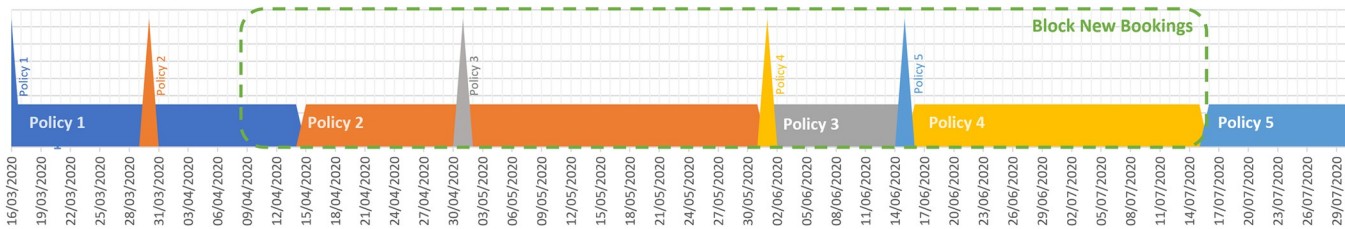

**Fig 1. First five ECP phases in the UK and their timings.**

eligible for free cancellation. For instance, when Policy 2 was set out on 30th March, the bookings covered by Policy 1 was still eligible until 14th April. This requires a method to distinguish the impact of each Policy by recognising its authoritative period without being interfered with by others valid during the same time.

The last reason that this period is particularly interesting is because of the mixture of additional Policies. When the cancellation Policy was first put forward, new bookings continued to go ahead. On 9th April 2020, however, Airbnb blocked booking for non-essential reservations under the accusation of tolerating irresponsible anti-lockdown behaviours [16]. This restriction lasted until the 15th of July for Scotland (https://spice-spotlight.scot/2022/11/25/timeline-of-coronavirus-covid-19-in-scotland/). The double effect of the two policies brought an opportunity to estimate the phases of cancellations without too much confusion being caused by new bookings.

## Summary

There is an urgent need to develop methods to track the market activities on STR online platforms in more detail due to the limitations of existing open or commercial data products, and the lack of data provided by platforms themselves. The highly unusual circumstances of the early stages of the pandemic provide a valuable opportunity to test the approach we are proposing as there was a sequence of cancellation periods (ECPs) which partially overlap in their timing but very few new bookings. In the remainder of the paper, we describe a method built upon understanding the meaning of Airbnb's daily calendar updates. We then exploit the complex reactions to the different phases of ECPs to show how our method permits a better understanding of market activity.

## Data and methods

### Data

To achieve better data transparency, granularity and quality, we set out a new approach that provides researchers with more fine-grained data through daily web scraping of platforms, in this case Airbnb. The scraping exercise is carefully designed and deployed as an automated Python program. Data was collected and stored in compliance with UK copyright law. The codebase was developed by UBDC and is openly available on GitHub (https://github.com/urbanbigdatacentre/ubdc-airbnb/tree/master/README).

The scraping exercise is fairly resource-intensive and the resulting data are unstructured text streams. Such extensive data contains consumer-facing information about listings (e.g. locations, property, amenities, nightly rates, booking calendars, ratings), hosts (e.g. hosting history, status, and ratings), and reviews (e.g. reviewer, time and contents for customer comments). This acts as a potential barrier, preventing many researchers from accessing the data, especially to the booking calendars, by themselves. To overcome these challenges, we derive a series of market indicators from the data aiming to: (a) clean, extract and encapsulate detailed changes on listing availabilities; (b) estimate the potential bookings/cancellations from the blocking/opening of days in the calendars; (c) estimate rental income taking into account nightly rental rates and additional fees; and (d) estimate visit durations and hence visitor numbers. These indicators retain listing activities to the most detailed extent while greatly reducing the size of the scraped data and enabling it to be readily analysed.

### Airbnb calendar and booking activities

Booking activities help us estimate occupancy rates and understand market activity levels. A booking calendar is attached to the listings every time we scrape information from the Airbnb

API. It contains the planned availability of a listing for each day over the following year. Hosts update their listing's availability as often as needed. InsideAirbnb provides these calendars approximately monthly while AirDNA provides a summary of listing performance built from daily calendars (including estimated occupancy rate and revenue https://www.airdna.co/vacation-rental-data/), but not the raw daily information.

The larger the time gap between collecting calendars, the less we know about activity levels. Bookings which are made and take place between two collection points would be missed completely, leading to undercounting while late cancellations may be missed, leading to over-estimation. On the other hand, the more often the calendars are scraped, the more complex the data is to process and manage. A more efficient data structure, retaining updates but also simplifying the calendar, is introduced next. We then describe a set of indicators using this data structure that estimate different aspects of market activities.

**Calendar activities–understanding of updates on availabilities.**   At the time of scraping, every listing shows its availability for the next 365 days in a booking calendar. Its status on each day is either 'available' or 'unavailable'. The latter covers days when a property is booked but also those when the host has taken it off the market for other reasons. One way we can use these data is to provide the core measure of interest to most researchers and policymakers which is the likely *occupancy levels* for listed properties. This offers two advantages over monthly scraping. First the latter only lets us observe whether a property was available or not on the date of the most recent prior scraping. With daily scraping, we observe its status on the day in question so we do not miss the impact of either late bookings or late cancellations. Second, we can use information on its status over all previous scrapings to get better information on likely bookings. We still cannot distinguish actual bookings from dates when the property was unavailable for other reasons but we can at least identify whether a given date was ever available and limit our count of bookings to occasions when the status changed from available to unavailable. The 'never available' statuses are therefore removed.

A second way to use the same data is to provide information on likely future activity levels for the year ahead of the day of scraping by tracking *bookings* and *cancellations* for future dates. Examining the updates to the booking calendar, we can aggregate the updates to the 365 days ahead of a particular scraping day. Although it is not a measure of the number of visits that will happen in practice, this does allow us to monitor how hosts adjust their availabilities in response to particular events such as a major conference or music festivals, or how the market is affected by sudden shocks such as the recent COVID-19 pandemic.

**Data structure for modelling calendar activities.**   With two statuses recorded for any given date for any property, we can observe four possible status changes between two scrapings (Fig 2). The listing can change from one state to the other or remain in either state. Fig 2 notes the potential ambiguities which result.

Based on this understanding, we summarize the data structure we used to retrieve calendar activities from daily scrapes for each listing (Fig 3). Starting from a given scraping date, we observe the calendar of bookings for the next 365 days on each day we scrape that listing, shown in Fig 3(A) with a value of '1' representing 'unavailable' and '0' available. The values for change between two successive days can also be represented as a two-dimensional array indexed by calendar dates and date of scraping (Fig 3(B)). The values are: '+1' when the availability on the same calendar date changed from available to unavailable; '-1' for a change from unavailable to available; or '0', unchanged.

This data structure allows us to derive the two measures of interest. To estimate *occupancy* on a given day, we use data from the row for that calendar date in (B). If the last non-zero value was '-1', we regard this as not occupied. If the last non-zero value was '+1', we regard this as occupied. If all values are zero, the listing may be either available or unavailable but we have

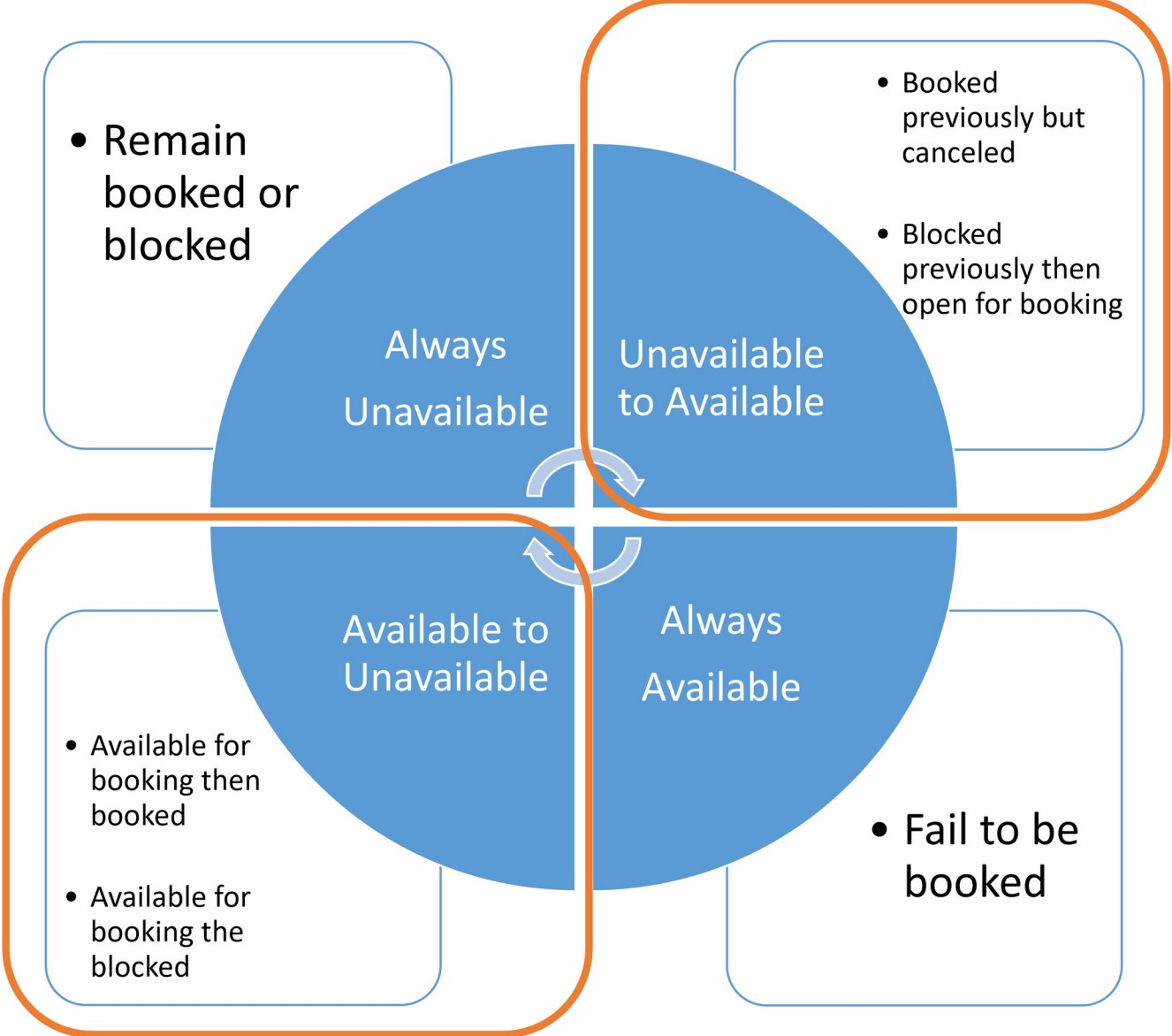

**Fig 2. Four types of booking updates on listing calendars.**

not observed a transition from available to unavailable so we do not regard this as occupied. The importance of daily scrapes here is that we do not miss status changes as we might do with, say, monthly records.

To examine future *bookings and cancellations* at a given date, we use data from the column for that date in (B). We count separately the number of '+1s', meaning days being booked, and '-1s', meaning cancellations to capture the volume of each kind of change so we can look at both absolute activity and net change in bookings. Again, the value of daily scraping is that we get a fine-grained (day-by-day) picture of activity volumes and we do not miss cases where properties might have bookings made and cancelled between, say, monthly scrapes. Whatever angle we take, our collected observations are subject to previously discussed limitations, most importantly, that we cannot distinguish dates which are booked from those otherwise unavailable.

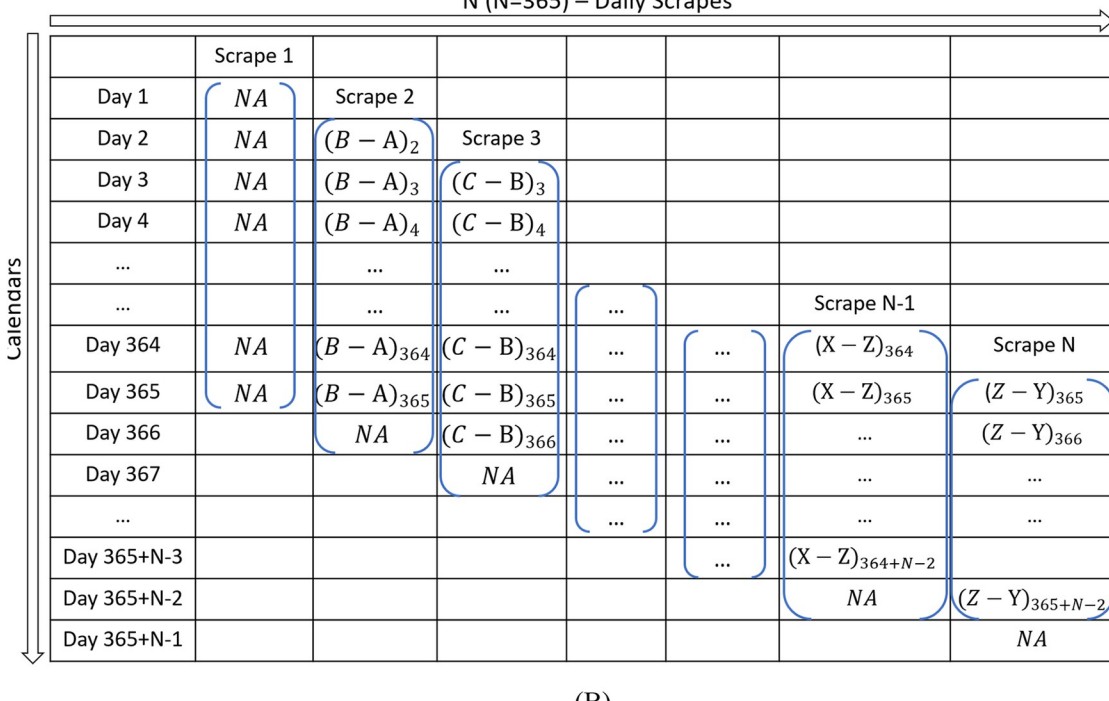

**Fig 3. Calendar activities defined from daily scrapings.** (A) Observations of daily booking calendars about a listing. (B) Calendar updates based on daily booking observations. For demonstration purpose, we assume N = 365, representing a years of scraping.

One limitation of our approach is that the method is somewhat vulnerable to discontinuities in scraping. These can occur through problems with the researcher's system or changes in the platform which require code to be updated. As we are comparing listings on successive days, the loss of one day's scraping means the loss of two days of changes. We were in the process of establishing our scraping during the early days of the pandemic, so lost a relatively larger number of days in the early part of that period, as is apparent in some of the results below. We could reduce the impacts of discontinuities by imputing data for missing days (e.g. at its simplest, by assuming no change and rolling forwards calendars to fill gaps). For the work here, we want to present our results with as little intervention as possible so we do not impute at any point.

For comparison, we compare the results using daily scrapes with those that would be obtained by weekly or fortnightly scraping. Due to the relatively short time periods for the different Policies, we cannot make results using monthly comparisons. For weekly or fortnightly scraping, the approach is the same as with daily but we make comparisons of the status of properties using wider intervals. The risk of course is that multiple changes within a period of time (e.g. booking and cancellation) may be missed completely.

**Estimations of rental income, visit duration and visitor numbers.**   The previous data structures help us estimate potential occupancy and bookings/cancellations daily. This lays the ground for estimation of revenue generated by each listing taking the rental price into account. This step helps planners and regulators to understand the likely scale of activity in an area, potential incentives to operate on the STR market and the potential levels of tax income or evasion.

The flexibility in Airbnb's business model allows hosts to adjust the nightly rental price every day. This nightly rate is recorded in the scraped calendar in addition to availability. As we do not know the cost to run the lettings, the value can be interpreted as *revenue* or *gross income*. Depending on the two ways of using the calendar updates, revenue gives us either a monetary return when we estimate occupancy or a potential income/loss when we estimate from the future calendar perspective.

Apart from the nightly rental income, hosts can charge a cleaning fee on top of every stay or visit. This money stream might be an additional income for the hosts. Unfortunately, Airbnb calendars do not distinguish bookings made by different users so we cannot identify how many such fees are levied. We define a *visit* as a group of consecutive days that are updated together in one scraping day. The number of consecutive days is identified as the *length of the visit*. The estimated number of visits also helps to capture the volume of the accommodated guests. This may help inform regulators about any changes in booking behaviour as a result of policy interventions. As two back-to-back bookings made on the same day might be incorrectly classified as one, this will tend to overstate visit durations and understate unique visitor numbers to a limited degree.

## Estimating the impact of ECPs in Edinburgh

### Airbnb in Edinburgh

To demonstrate the value of our methods, we apply them to the Airbnb market in Edinburgh, tracking reactions to the company's interventions in response to the COVID-19 pandemic. Edinburgh is the second most popular tourist city in the UK and Airbnb has a strong presence [47] with over 10,000 active listings in May 2019 [30]. In the central city where the sector is most concentrated, 79% of listings are rented as entire homes, accounting for one-in-six (16%) of all dwellings. We focus on entire home rental listings because they are of primary concern for regulation.

The rapid growth and high spatial concentration have caused concerns for community groups and the local authority. Under the recent STR regulations introduced in Scotland, Edinburgh has set the whole city as a control area. All listings will be required to have a licence by July 2024 [48]. To understand levels of compliance and balance its economic contributions, the local authority will need detailed and up-to-date tracking of the sector.

Our scraping exercise accumulated data on 10,489 listings in Edinburgh between January and July 2020, with 63% categorized as entire home rentals. These figures align well with the recent Scottish Government report on the sector using InsideAirbnb data for May 2019 which reported 9994 Airbnb listings, of which 66% were entire home rentals [30].

### Tracking cancellations in Edinburgh under the ECPs

To demonstrate the capability of our method, we start by examining the cancellations made under each Policy (Table 1). Cancellations under a given Policy can be made from the date the Policy was announced up to the last day of the eligible period (the 'Coverage Days' in Table 1). On many dates, cancellations could be made under different Policies. We identify the relevant Policy from the dates of the bookings which are cancelled giving a count of cancelled bookings for each date under each Policy. The 'Observed Days' in Table 1 is the number of days for which we have scraping records covering successive days in order to measure cancellations on

**Table 1. Summary of indicators.**

| | | Policy 1 | Policy 2 | Policy 3 | Policy 4 | Policy 5 |
|---|---|---|---|---|---|---|
| Announced at | | 2020-03-16 | 2020-03-30 | 2020-05-01 | 2020-06-01 | 2020-06-15 |
| Eligible Period | | 2020-03-16, | 2020-04-15, | 2020-06-01, | 2020-06-16, | 2020-07-16, |
| | | 2020-04-14 | 2020-05-31 | 2020-06-15 | 2020-07-15 | 2020-07-31 |
| Coverage Days | | 30 | 61 | 45 | 45 | 47 |
| Observed Days | | 3 | 18 | 19 | 34 | 45 |
| Cancelled Bookings (Nights) | | | | | | |
| | Total | 12429 | 7843 | 4949 | 8080 | 10900 |
| | Average | 4143 | 435.7 | 247.5 | 237.6 | 242.2 |
| | Median | 4098 | 320.5 | 228 | 191. | 249 |
| | Max | 4702 | 1274 | 891 | 908 | 792 |
| | Min | 3629 | 13 | 1 | 6 | 9 |
| Cancelled Bookings (Revenue) | | | | | | |
| | Total | £1,136,204 | £780,739 | £500,695 | £971,204 | £1,350,297 |
| | Average | £378,735 | £43,374 | £25,035 | £28,565 | £30,007 |
| | Median | £360,738 | £28,344 | £23,583 | £22,179 | £26,081 |
| | Max | £442,703 | £118,729 | £78,238 | £145,657 | £126,471 |
| | Min | £332,763 | £1,240 | £100 | £1,019 | £1,318 |
| No. Visits | | | | | | |
| | Total | 3754 | 1973 | 1182 | 1507 | 2437 |
| | Average | 1251.3 | 109.6 | 62.2 | 44.3 | 54.2 |
| | Median | 1182 | 68.5 | 61 | 40 | 54 |
| | Max | 1454 | 276 | 186 | 129 | 148 |
| | Min | 1118 | 1 | 2 | 5 | 3 |
| Median Length of Visit | | | | | | |
| | Average | 3 | 2.3 | 2.9 | 3.1 | 2.4 |
| | Median | 3 | 3 | 3 | 3 | 3 |
| | Max | 3 | 3 | 3.5 | 7 | 4 |

that date. As noted previously, discontinuities in scraping, particularly in the early days of the pandemic, mean that Observed Days can be much fewer than Coverage Days (daily tracking are shown in S2, Fig S2.1 in S1 Appendix).

Methodologically, we account for the impact of a Policy by locating the daily scrapes (columns of Fig 3B) for the days from when the Policy was first announced up to the last date of the eligible period. We then select the booking dates (rows of Fig 3B) indexed by the dates eligible for cancellation STR under each Policy. For each scraping date, we count the number of cancelled nights made on that day (count the number of '-1s') and derive the lost revenue by applying the known charge for the relevant date. Where consecutive booking days are cancelled on the same date, we treat these as a single visit and hence derive number of visits and length of visits. As noted already, there is obviously a risk that two or more consecutive bookings may be cancelled on the same date although we think this is likely minor but our estimate of visit length should be treated as an upper limit.

Table 1 presents the derived indicators, including cancelled bookings and visits, under the various Policies. It shows that bookings were cancelled particularly rapidly during Policy 1, the first ECP of the pandemic. The high rates reflect the fact that the opportunity to cancel was only given at the start of the affected period rather than ahead of it and covered a period when bookings would have been made at normal (pre-pandemic) rate. With later Policies, fewer bookings would have been made because of the uncertainties around restrictions on movement and there was more time to make cancellations ahead of the booking period covered. Cancellations are therefore more dispersed under later Policies. The average and median length of visits cancelled are very similar, however. Across all five ECPs, just over 10,900 visits with 44,000 nights of bookings were cancelled, representing £4.7m in revenue.

Fig 4 shows the number of days (booking nights) cancelled on each calendar day, broken down by the Policy to which they apply. The number of cancellations per day gradually decreases over time, with the peak cancellation rate occurring at the beginning of Policy 1 when ECPs were first introduced.

Fig 5 shows the number of cancelled visits made on each date under each Policy, along with visit lengths. Cancelled visits show a gradual decrease over time, in line with previous trends (Fig 5, left axis). However, for Policy 4, there is a higher potential revenue loss than Policy 2 despite a lower level of cancelled visits. Our method makes clear the reason for this which is the greater length of cancelled visits (Fig 5, right axis). Policy 4 covers the bookings made for mid-June to mid-July which is the start of the tourism season and normally piled up with events and festivals. Visits in Policy 5 are likely shorter than usual because enhanced cleaning periods were introduced to prevent cross-contamination [49] while another round of rising infection rates started around that time [50].

## Tracking other market activity during the ECP period

In general, the STR market was closed during the period studied here, at least until mid-July, but there were still some activities captured. As bookings were cancelled and dates became 'available', hosts could change them to 'unavailable' to prevent further bookings. They might also return the date to 'available' if they were open to bookings by 'essential workers' which were permitted during this time. Towards the end of the lockdown period, they may also have reopened for bookings in the hope that conditions might permit travel by the time of the booking.

Our method makes these activities visible. In Table 2, we look at later activity on properties for the dates where bookings had been cancelled (i.e. those captured in Table 1). In the early stages of the pandemic, hosts appear to have simply left bookings 'available' since it was clear

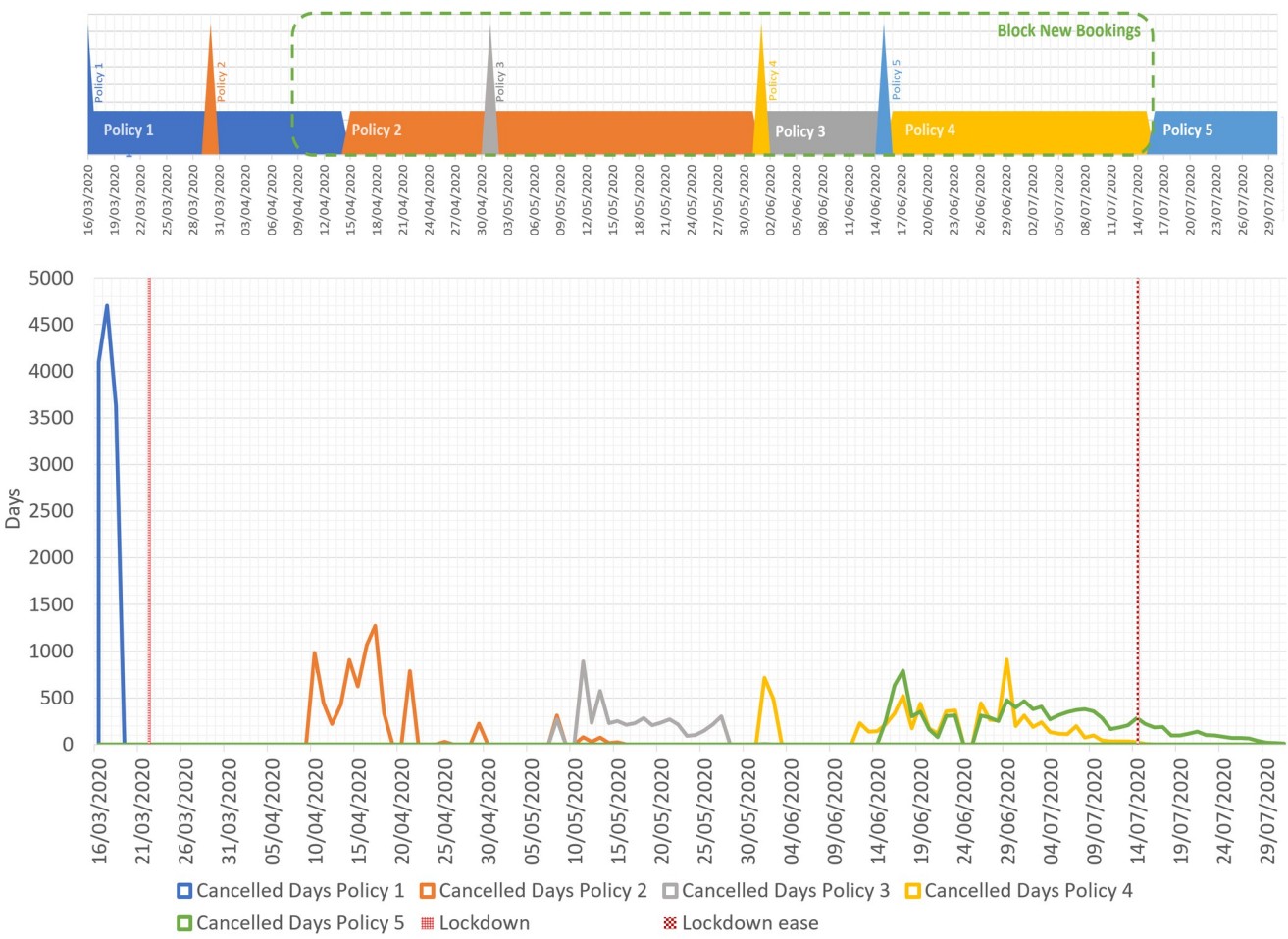

**Fig 4. Cancellation in days and estimated cancelled revenue by date of cancellation.**

no visits were possible (94.8% of cancelled booking dates in Policy 1 saw no further updates, for example). As time went on, owners were more likely to alter the status to mark properties as 'unavailable' with the percentage rising from 5% to 66%. Only a very small proportion saw any additional activity beyond these changes, and this was more prevalent later on in this period.

On the other side of the market, there are listing dates which had not been booked at the start of the pandemic where the availability of days changes from available to unavailable. In normal market conditions, we would view this as indicating bookings but in this special period, we regard them as dates are being blocked by hosts. Table 3 focuses on these dates and shows that, in the great majority of cases (91% under Policy 4 to 97% under Policy 1), the booking remained blocked. Very small proportions were subsequently switched back to available, and again this is possibly opening for key workers. The volumes of re-opening and subsequent booking rose under Policies 4 and 5.

## Comparing daily with weekly and fortnightly scraping methods

A key aim of our work is to demonstrate the value of daily scraping in providing a fine-grained picture of activity in the STR market. One further way to demonstrate this is through a comparison with measures which would be provided by less frequent scraping. To explore this, we

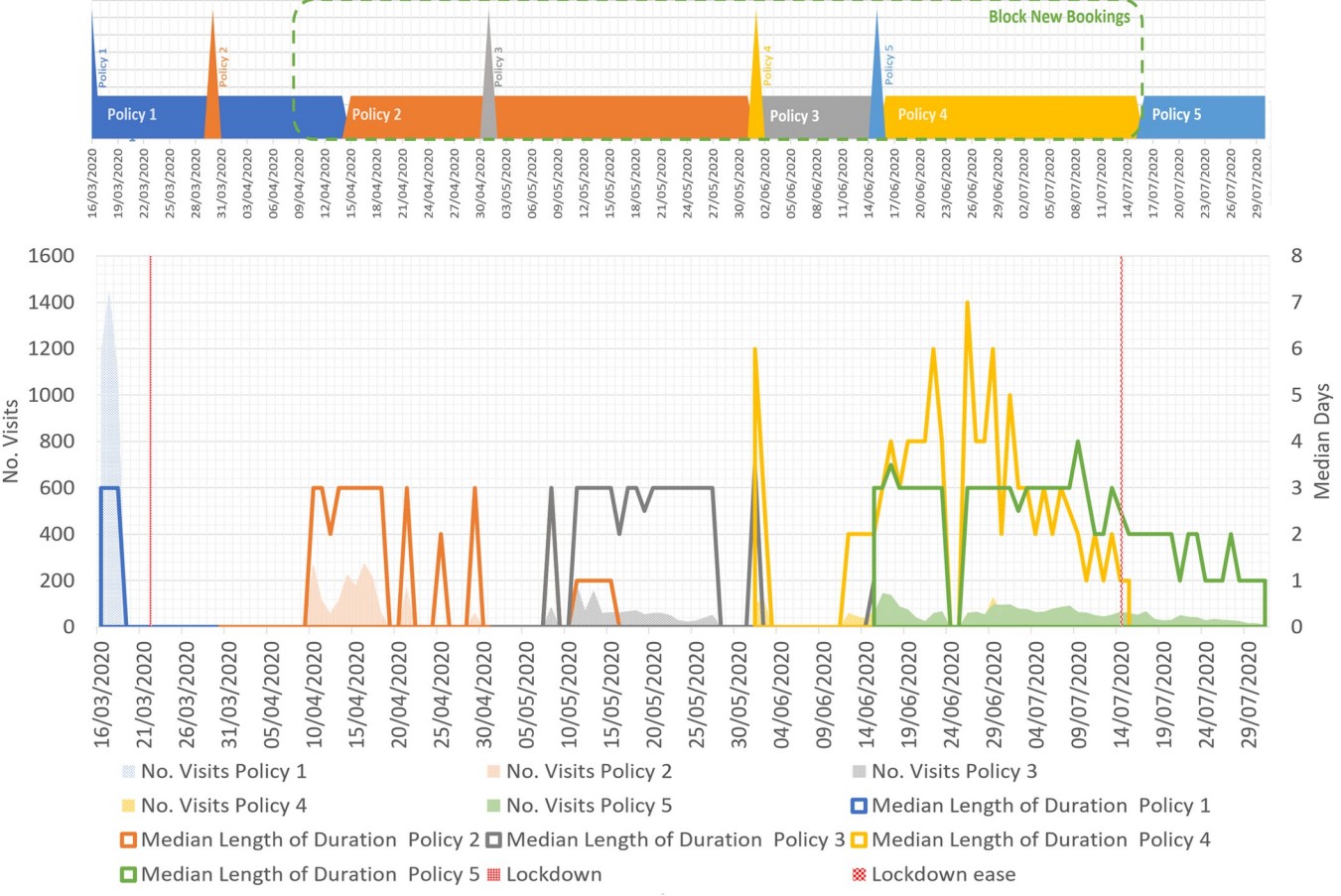

**Fig 5. Potential visits were being cancelled and median length of cancelled visits.**

compared patterns from daily methods with those from weekly and fortnightly updates. Monthly patterns could not be evaluated since the ECP periods were relatively short, as noted above.

Less frequent scraping leads to very different estimates of Policy impacts and greater sensitivity to data gaps (Table 4). Notably, we observed significant differences in estimates of the total number of cancelled days. With weekly measures, there are significant fluctuations in estimates–some higher, others much lower than suggested by daily scraping. The higher weekly estimations were influenced by data breaks that occurred during the initial setup of scraping in the early stages of the policies. When comparing two calendars within a weekly sampling interval, the presence of both sets of data becomes more likely. With fortnightly

**Table 2. Further calendar updates to cancelled days.**

|  | Total cancelled | No further updates | Update to unavailable Once | Multiple Updates |
|---|---|---|---|---|
| Policy 1 | 12429 | 94.8% | 5.1% | 0.1% |
| Policy 2 | 7843 | 65.6% | 32.0% | 2.4% |
| Policy 3 | 4949 | 47.0% | 44.9% | 8.1% |
| Policy 4 | 8080 | 27.6% | 61.2% | 11.2% |
| Policy 5 | 10900 | 25.5% | 66.4% | 8.0% |

**Table 3. Further calendar updates to blocked days.**

|  | Total Blocked | No further Updates | Updated to Available Once | Multiple Updates |
|---|---|---|---|---|
| Policy 1 | 8790 | 97.1% | 2.8% | 0.1% |
| Policy 2 | 53213 | 97.1% | 1.1% | 1.8% |
| Policy 3 | 21649 | 95.9% | 1.5% | 2.6% |
| Policy 4 | 34405 | 91.6% | 1.3% | 7.0% |
| Policy 5 | 45398 | 92.4% | 1.9% | 5.7% |

measures, all suggest a substantially lower level of activity than captured by daily scraping with more than half missed for every Policy. The findings highlight the potential influence that varying data quality could have on such analyses.

## Conclusions and discussion

The contribution of this paper is to show the importance of focussing on data quality and methods for the study of the platform economy for short-term accommodation. We have used the experiences of one city during a highly unusual period to illustrate this. There is a growing body of literature on the scale and nature of the STR sector but almost all the quantitative work relies on data provided by third parties, either commercial (e.g. AirDNA) or third-sector (e.g. InsideAirbnb). Methods for the former are opaque while for the latter they are limited by resource constraints to infrequent collection of data. Little attention is paid in the literature to the impacts of data sources and quality on our understanding of the scale of activity in this sector.

We show how daily scraping can be used to provide more accurate and temporally detailed estimates of booking and cancellation activity, and hence of occupancy. Combining nightly rental costs with occupancy provides estimates of income while patterns of daily changes provide estimates of visit length and hence visitor numbers. The use of data with infrequent collection periods may lead to a significant under- or over-estimation of market activity. The larger collection intervals also make the estimations more sensitive to data breaks. The current data providers rarely make clear their scraping quality which may significantly affect analysis results.

There are of course a number of assumptions still required to arrive at these estimates but they are far fewer than required when working with less fine-grained data. There is further work which could be done to refine our approach. The code for collecting data was in its early stages at the start of the period examined here and we suffered some breaks in collection which impact some results. For instance, there is a data gap of approximately two weeks between Policy 1 and Policy 2. This gap may lead to an underestimation of the true impact of these policies. Nevertheless, our scraping process effectively captures the surge in cancellations resulting from the unprecedented market interventions when Policy 1 was put forward. As the scraping

**Table 4. Total cancelled days estimated from daily, weekly and fortnightly scraping methods.**

|  | Policy 1 | Policy 2 | Policy 3 | Policy 4 | Policy 5 |
|---|---|---|---|---|---|
| Daily | 12429 | 7843 | 4949 | 8080 | 10900 |
| Weekly(Monday) | 10419 | 9608 | 2491 | 8867 | 8690 |
| %Daily | 83.8% | 122.5% | 50.3% | 109.7% | 79.7% |
| Fortnightly(Monday) | 1009 | 3250 | 2203 | 1463 | 4598 |
| %Daily | 8.1% | 41.4% | 44.5% | 18.1% | 42.2% |

process becomes more consistent, minor data gaps—such as those lasting only a couple of days during Policy 2 and Policy 3—are likely to exert a smaller influence on the overall estimation, particularly when we employ interpolation techniques like rolling averages. The greater the stability of the scraping process, the more precise and robust our estimations become. While we have greatly improved the resilience, we will need to develop methods to cope with missing data through imputation. Such imputation on calendar updates requires accessing the historical records of availability status changes. The absent data can be inferred from the surrounding availability changes, particularly when a single visit is booked for multiple days. We have not implemented these here since we wanted to show results with least manipulation but we have already begun to implement some basic approaches in subsequent work.

More generally, this work highlights the necessity for policy makers to think about data needs when regulating STR market. STRs have substantial impacts on cities yet there is still a lack of detailed data sources for research and regulation purposes. Despite their size and complexity, databases of daily booking calendars collected via scraping are particularly important in understanding key market trends such as occupancies, income, visit durations and visitor numbers. The web scraping code (https://github.com/urbanbigdatacentre/ubdc-airbnb/tree/master/README) and data methods (https://github.com/YangWang-Glasgow/Airbnb-Processing-Daily-Booking-Calendars) have been openly available. They are extensible to other cities worldwide, effectively addressing both regulatory and research requirements.

Some limitations are worth noting. Technically, our scraping process accesses consumer-facing web content daily through the current Airbnb API since 2020. A vigilant monitoring of website responses to the scraping approach is needed, and our method and abstract data structure are designed to easily accommodate future changes. The most pertinent is the continuing ambiguity of the 'unavailable' status in the booking calendar. Our method is designed to reduce this by counting a booking only where the status is observed to change from available to unavailable but it cannot remove the possibility that this is a host blocking dates rather than a booking. This likely results in some overestimation of occupancies. A requirement for transparency from platforms here would assist research and regulation.

Finally, our suggestion to those who want to use these trends is that, although indicators are designed to reflect market activities, they are not self-explanatory and need to be interpreted in context. Trends in a given city may respond to particular local regulations or housing laws as well as national or global events. The proposed methods are intended to support a wider investigation of the complex factors driving STR market demand and supply, and of the impacts of the sector on the host cities. The detailed daily tracking of calendar activities will provide extra insights into the life cycle of listings and lay the ground for better monitoring of the changing market landscape.

## Supporting information

**S1 Appendix.**
(DOCX)

**S1 File.**
(XLSX)

## Acknowledgments

We thank data scientist, Mr Nick Ves, for setting up the daily scraping exercises at Urban Big Data Centre.

## Author Contributions

**Conceptualization:** Yang Wang.

**Formal analysis:** Yang Wang.

**Funding acquisition:** Mark Livingston, David P. McArthur, Nick Bailey.

**Investigation:** Yang Wang.

**Methodology:** Yang Wang, Nick Bailey.

**Software:** Yang Wang.

**Validation:** Yang Wang, Mark Livingston, Nick Bailey.

**Visualization:** Yang Wang.

**Writing – original draft:** Yang Wang.

**Writing – review & editing:** Yang Wang, Mark Livingston, David P. McArthur, Nick Bailey.

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
