## [Decision Letter · Decision Letter 0]

4 Sep 2023

PONE-D-23-19176Enhancing our Understanding of Short-term Rental Activity: A Daily Scrape-based Approach for Airbnb ListingsPLOS ONE

Dear Dr. Wang,

Thank you for submitting your manuscript to PLOS ONE. After careful consideration, we feel that it has merit but does not fully meet PLOS ONE’s publication criteria as it currently stands. Therefore, we invite you to submit a revised version of the manuscript that addresses the points raised during the review process.

 The manuscript is structured and well-written. However, some issues need attention and revision before publication, particularly the variable description and result presentation. A minor revision is recommended for this submission. ==============================

We look forward to receiving your revised manuscript.

Kind regards,

Sutee Anantsuksomsri

Academic Editor

PLOS ONE

2. In your Methods section, please include additional information about your dataset and ensure that you have included a statement specifying whether the collection and analysis method complied with the terms and conditions for the source of the data.

“This work was supported by ESRC-funded Urban Big Dat Centre (UBDC) [ES/L011921/1 and ES/S007105/1].”

Additional Editor Comments:

The manuscript is structured and well-written. However, prior to moving forward with publication, there are some issues that require attention and revision. A minor revision is recommended for this submission.

Reviewers' comments:

Reviewer's Responses to Questions

**Comments to the Author**

1. Is the manuscript technically sound, and do the data support the conclusions?

Reviewer #1: Yes

Reviewer #2: Yes

2. Has the statistical analysis been performed appropriately and rigorously? 

Reviewer #1: N/A

Reviewer #2: Yes

3. Have the authors made all data underlying the findings in their manuscript fully available?

Reviewer #1: Yes

Reviewer #2: Yes

4. Is the manuscript presented in an intelligible fashion and written in standard English?

Reviewer #1: Yes

Reviewer #2: Yes

5. Review Comments to the Author

Reviewer #1: The paper presents a novel method for data collection of STR platform (Airbnb) with the focus on the fine granularity of occupancy information. The contribution is clear and significant. The manuscript is well written and easy to follow.

However, there are a few minor points of improvement. First, the author should explicitly define the variables extract from scrapping. The author discussed the daily occupancy and price, but it made me wonder if other variables like types of accommodations, locations (neighborhood), or characters of hosts were extracted from this method of web scraping. It would be helpful to provide a list of variables.

Second point involves the presentation of results. Timing of each policy as shown in Fig. 1 should be include in the results. In other words, Fig. 4 and 5 should be overlaid with Fig. 1 so that the time lag in cancelation can be visually analyzed in accordance with the timeframe of the policy.

The last point is about the consistency in the in-text citation, specifically, in line 58 of p.3.

If these points are addressed by the author, the manuscript would be in a better shape for publication.

Reviewer #2: The paper titled "Enhancing our Understanding of Short-term Rental Activity: A Daily Scrape-based Approach for Airbnb Listings" presents an interesting and valuable approach for studying short-term rental (STR) activity, specifically focusing on Airbnb listings. The authors address the challenge of data scarcity and quality in studying the impact of the online short-term rental market on housing supply and policy interventions. The paper proposes a methodology involving daily web scraping of Airbnb calendars to derive fine-grained data, aiming to provide better insights into booking activities, occupancy rates, and market trends. The methodology is demonstrated through its application to the Airbnb market in Edinburgh during the COVID-19 pandemic.

Strengths:

Original Approach: The paper's approach of daily web scraping to collect detailed data from Airbnb calendars is a novel and innovative way to address the limitations of existing data sources. This approach can potentially offer a more accurate and temporally detailed understanding of STR activity, which is crucial for policy evaluation and market analysis.

Relevance: The topic of short-term rentals and their impact on housing supply and urban policies is highly relevant, especially in the context of platforms like Airbnb. The paper's focus on providing data-driven insights to aid policy interventions is important for informed decision-making.

Data Transparency: The authors emphasize the importance of data transparency and make their codebase openly available on GitHub, enhancing the reproducibility and credibility of their research. This practice promotes collaboration and peer review within the research community.

Real-world Application: The application of the proposed methodology to analyze the effects of "Extenuating Circumstances Policies" (ECPs) during the COVID-19 pandemic provides a practical demonstration of the approach's value. The case study in Edinburgh adds real-world context to the research.

Weaknesses:

-Data Quality and Reliability: The paper emphasizes data quality but does not thoroughly address issues related to data breaks, missing data, and the potential impact of data quality on the analysis. Discussing potential sources of bias and limitations due to data quality would enhance the paper's rigor.

Please discuss about:

>>Describe the strategies implemented to address or mitigate issues related to data quality, including handling data breaks and missing data. This could involve discussing imputation methods or data validation procedures.

>>Present a clear analysis of the potential impact of data quality issues on the results. This could involve sensitivity analyses that assess how variations in data quality might affect key findings.

-Platform and Contextual Changes: The study focuses on a specific time period (COVID-19 pandemic) and a specific location (Edinburgh). The discussion should include considerations about the generalizability of findings to other time periods, locations, and different regulatory contexts.

Please discuss about:

>>Address the potential influence of platform changes (such as modifications to Airbnb's calendar format) on the data collection and analysis process. Explain how the proposed methodology adapts to such changes and maintains its accuracy.

>>Discuss the transferability of the approach to different locations and time periods, along with potential challenges and adjustments required to apply the methodology in diverse contexts.

Conclusion:

The paper presents a valuable approach to address the limitations of existing data sources in studying short-term rental activity, particularly in the context of Airbnb. The methodology's potential benefits for policy evaluation and market analysis are clear. However, to strengthen the paper, it is essential to address the weaknesses mentioned above. More in-depth discussions of the methodology's limitations, data quality issues, and the broader applicability of findings would provide a more comprehensive understanding of the approach's strengths and limitations. Additionally, further explanation of assumptions, potential biases, and strategies to handle missing data would enhance the paper's credibility and usefulness to both researchers and policymakers.

6. PLOS authors have the option to publish the peer review history of their article (what does this mean?). If published, this will include your full peer review and any attached files.

Reviewer #1: No

Reviewer #2: **Yes: **Wasit Limprasert

---

## [Author Response · Author response to Decision Letter 0]

14 Dec 2023

Yang Wang

Urban Big Data Centre, 

University of Glasgow

7 Lilybank Gardens, Hillhead, 

Glasgow G12 8RZ 

11 October 2023

To: Prof. Sutee Anantsuksomsri

Academic Editor

PLOS ONE

Dear PLOS ONE Editor Prof. Sutee Anantsuksomsri,

We wish to submit the revised research article entitled “Enhancing our Understanding of Short-term Rental Activity: A Daily Scrape-based Approach for Airbnb Listings” based on well-appreciated comments and suggestion made by PLOS ONE editor and reviewers.

In this paper, we propose a systematic method to capture, manage and interpret short-term rental activities derived from daily scrapings of the digital platform, Airbnb. This is significant because the existing data sources, whether proprietary or open, do not provide the required detail and rigour needed to understand the performances of such businesses which are found to exert pressure on cities’ affordable housing. The regulation, in many cities, urgently needs to balance between harnessing their potential contributions to the local economy and mitigating their adverse effects on neighbourhoods and communities.

We deeply appreciate both the editor and reviewers for your invaluable feedback. Your insights have been instrumental in refining our paper. We would like to confirm that we've addressed all the issues and questions raised about the content. We've submitted a marked-up copy titled 'Revised Manuscript with Track Changes' to highlight the modifications from the original version. Additionally, we've provided an unmarked version labelled 'Manuscript', which has been formatted according to PLOS ONE guideline.

We have made responses to the constructive suggestions in ‘Responds to Reviewers’ along with that we have highlighted corresponding changes made on the revised manuscript. 

We fully agree with PLOS ONE’s dedication to openness in research methods and data sharing. Under the UK copyright law, we cannot share the raw data. We open the codebase for web scraping at https://github.com/urbanbigdatacentre/ubdc-airbnb/tree/master/README and the method for calculating the indicators along with the aggregate tables for this paper are open at https://github.com/YangWang-Glasgow/Airbnb-Processing-Daily-Booking-Calendars. The figures that have been used to produce the charts and plots in the paper are also provided in the Supporting Information files.

We confirm that this work is original and has not been published elsewhere, nor is it currently under consideration for publication elsewhere.

We have no conflicts of interest to disclose. 

Finally, we want to confirm that this work has funding support from ESRC (ESRC-funded Urban Big Dat Centre (UBDC) [ES/L011921/1 and ES/S007105/1]). The funders had no role in study design, data collection and analysis, decision to publish, or preparation of the manuscript.

Please address all correspondence concerning this manuscript to me at yang.wang@glasgow.ac.uk.

Thank you for your suggestions and comments again. 

We hope the revised paper now satisfies the requirements of PLOS ONE.

Yours Sincerely,

Yang Wang

Research Associate

Urban Big Data Centre, University of Glasgow

--point by point response to reviewers

Response to academic editor:

We are very grateful to the reviewers and the editor for their careful and constructive feedback. We have responded to all the points raised below and believe that the paper is significantly strengthened as a result. 

Authors response: Thanks very much for suggesting this! We have updated the format of the paper according to the guidance. 

2. In your Methods section, please include additional information about your dataset and ensure that you have included a statement specifying whether the collection and analysis method complied with the terms and conditions for the source of the data.

Authors response: We have added ‘Data was collected and stored in compliance with UK copyright law’ in line 254 (Page 10).

“This work was supported by ESRC-funded Urban Big Dat Centre (UBDC) [ES/L011921/1 and ES/S007105/1].”

Authors response: We have added ‘Funding support is from ESRC-funded Urban Big Data Centre (UBDC) [ES/L011921/1 and ES/S007105/1]. The funders had no role in study design, data collection and analysis, decision to publish, or preparation of the manuscript’ in the acknowledgement in paper line 566 (Page 23). Title page and cover letter are also updated accordingly.

Authors response: We wholeheartedly appreciate the journal's dedication to openness in research methodologies and data sharing, as we believe this approach greatly benefits the broader research community. In our paper, we propose an open methodology that critically examines short-term rental market data obtained by employing web scraping techniques on the Airbnb platform. Our primary objective is to demonstrate the value of this approach for researchers who are otherwise solely reliant on proprietary data or open data which are much more limited in scale (see paper for details).

Our scraping exercise accesses the openly-available Airbnb listings data under the provisions of UK copyright law. In the UK, the text and data mining exemption to copyright law permits the collection, storage and analysis of data which researchers have a legal right to view (in this case, Airbnb’s public property listings) where the purpose is non-commercial academic research. However, the law does not permit the sharing or distribution of the raw data to others. For work in our field which uses this approach to data collection, it is therefore not possible for researchers to provide direct access to the raw data they have used. For a detailed discussion of the legal issues, please see: Burrow, S. (2021) The Law of Data Scraping: A review of UK law on text and data mining. CREATe Working Paper 2021/2 (https://doi.org/10.5281/zenodo.4635759).

Our team at Urban Big Data Centre has provided details of the methods and code used to collect the data and these are available to others so they can create their own data collections (details here: https://github.com/urbanbigdatacentre/ubdc-airbnb/tree/master/README). 

In light of our commitment to transparency, we had made two additional steps to ensure that our paper meets the journal's requirement of making minimal data fully available:

(a) We make the code used to process the data and generate our analyses accessible through GitHub [https://github.com/YangWang-Glasgow/Airbnb-Processing-Daily-Booking-Calendars]. This approach would provide complete transparency regarding our methods, allowing other researchers to scrutinize, repeat and build upon our work. It cannot support direct replication however since we cannot legally share the dataset used in the paper.

(b) Additionally, we share the aggregated data used in the creation of our figures as part of the Supporting Information files. This step would facilitate a understanding of our findings and enable other researchers to utilize this aggregated data for further analyses or validation.

We hope these proposed measures align with the journal's data availability requirements and they would be considered sufficient to ensure the minimal data necessary for replication and validation are fully accessible to the research community. We are unable to go further while remaining within the UK law. 

Authors response: We have thoroughly reviewed both the reference list and citations, ensuring that they also address the suggestions put forth by Reviewer 1. 

Response to Reviewer #1:

Reviewer #1: The paper presents a novel method for data collection of STR platform (Airbnb) with the focus on the fine granularity of occupancy information. The contribution is clear and significant. The manuscript is well written and easy to follow.

Authors response: Thank you very much for this positive comment. 

However, there are a few minor points of improvement. First, the author should explicitly define the variables extract from scrapping. The author discussed the daily occupancy and price, but it made me wonder if other variables like types of accommodations, locations (neighborhood), or characters of hosts were extracted from this method of web scraping. It would be helpful to provide a list of variables.

Authors response: Our data scraping exercises indeed collect a substantial volume of information, encompassing details about listings (e.g., locations, property attributes, amenities, nightly rates, booking calendars, and ratings), hosts (e.g., hosting history, status, and ratings), and reviews (e.g., reviewer information, timestamps, and customer comments). This data is stored in an unstructured text format synchronized with our server on a daily basis.

The most challenging aspect of analysing the market is gaining access to listings’ frequently updated calendars, a detail often omitted from existing datasets. As noted in our paper, AirDNA provides estimated occupancy rates, but their machine-learning algorithm lacks transparency. Their data is also not open or free to access. Additionally, data from InsideAirbnb is released monthly. It may result in underestimating the daily booking updates, a crucial feature of short-term rental platforms like Airbnb.

Our paper focuses on addressing this challenge by sharing our scraping codebase and proposing a method to extract booking and cancellation details from listing calendars, considering factors such as the number of days, potential income and losses, and the frequency of visits. We have also made the code developed for this paper openly available on the author’s Github[https://github.com/YangWang-Glasgow/Airbnb-Processing-Daily-Booking-Calendars] and aim to provide the broader research community access to both the data and methodology in this field. 

We have updated line 259 on page 10, ‘Such extensive data contains consumer-facing information about listings (e.g. locations, property, amenities, nightly rates, booking calendars, ratings), hosts (e.g. hosting history, status, and ratings), and reviews (e.g. reviewer, time and contents for customer comments). This acts as a potential barrier, preventing many researchers from accessing the data, especially to the booking calendars, by themselves’, detailing more information we collected from the scarping exercise. 

Second point involves the presentation of results. Timing of each policy as shown in Fig. 1 should be include in the results. In other words, Fig. 4 and 5 should be overlaid with Fig. 1 so that the time lag in cancelation can be visually analyzed in accordance with the timeframe of the policy.

Authors response: We have updated Fig 4 and Fig 5 accordingly to improve the readability of the paper.

The last point is about the consistency in the in-text citation, specifically, in line 58 of p.3.

Authors response: Thank you for bringing this issue to our attention. We have updated all the citations and have ensured that they now conform to a consistent reference style. 

Reviewer #2: The paper titled "Enhancing our Understanding of Short-term Rental Activity: A Daily Scrape-based Approach for Airbnb Listings" presents an interesting and valuable approach for studying short-term rental (STR) activity, specifically focusing on Airbnb listings. The authors address the challenge of data scarcity and quality in studying the impact of the online short-term rental market on housing supply and policy interventions. The paper proposes a methodology involving daily web scraping of Airbnb calendars to derive fine-grained data, aiming to provide better insights into booking activities, occupancy rates, and market trends. The methodology is demonstrated through its application to the Airbnb market in Edinburgh during the COVID-19 pandemic.

Strengths:

Original Approach: The paper's approach of daily web scraping to collect detailed data from Airbnb calendars is a novel and innovative way to address the limitations of existing data sources. This approach can potentially offer a more accurate and temporally detailed understanding of STR activity, which is crucial for policy evaluation and market analysis.

Relevance: The topic of short-term rentals and their impact on housing supply and urban policies is highly relevant, especially in the context of platforms like Airbnb. The paper's focus on providing data-driven insights to aid policy interventions is important for informed decision-making.

Data Transparency: The authors emphasize the importance of data transparency and make their codebase openly available on GitHub, enhancing the reproducibility and credibility of their research. This practice promotes collaboration and peer review within the research community.

Real-world Application: The application of the proposed methodology to analyze the effects of "Extenuating Circumstances Policies" (ECPs) during the COVID-19 pandemic provides a practical demonstration of the approach's value. The case study in Edinburgh adds real-world context to the research.

Authors response: Thanks so much for your positive comments.

Weaknesses:

-Data Quality and Reliability: The paper emphasizes data quality but does not thoroughly address issues related to data breaks, missing data, and the potential impact of data quality on the analysis. Discussing potential sources of bias and limitations due to data quality would enhance the paper's rigor.

Please discuss about:

>>Describe the strategies implemented to address or mitigate issues related to data quality, including handling data breaks and missing data. This could involve discussing imputation methods or data validation procedures.

Authors response: Thanks for this suggestion. This paper introduces a method that involves accessing Airbnb listings' booking calendars (Section 10), organizing calendar updates into an efficient data structure (Page 12), and estimating key performance indicators (Page 12- 14). Our primary focus lies in examining the data structure and comparing the benefits of obtaining daily scrapes versus other datasets with more extended scraping intervals (Page 20). Additionally, this methodology provides the capability to monitor various market activities through continuous tracking of changes in listing booking statuses, after bookings or cancellations have occurred (Page 18).

While we highlight the potential of this approach, we acknowledge in conclusion and discussion section (Page 21) that there are potential impact from missing data or data gaps. We expand on this discussion by referencing a case study in line 521. The absence of data may lead to an underestimation of listing performance. ‘For instance, there is a data gap of approximately two weeks between Policy 1 and Policy 2. This gap may lead to an underestimation of the true impact of these policies. Nevertheless, our scraping process effectively captures the surge in cancellations resulting from the unprecedented market interventions when Policy 1 was put forward. As the scraping process becomes more consistent, minor data gaps—such as those lasting only a few days during Policy 2 and Policy 3—are likely to exert a smaller influence on the overall estimation, particularly when we employ interpolation techniques like rolling averages. The greater the stability of the scraping process, the more precise and robust our estimations become.’

In line 530 (Page 22), we also discuss potential methods of imputation to address smaller gaps. ‘Such imputation on calendar updates requires accessing the historical records of availability status changes. The absent data can be inferred from the surrounding availability changes, particularly when a single visit is booked for multiple days.’ This approach enables us to obtain a reliable 'guess' of the final booking status by considering bookings as visits. We are cautious about implementing more hasty imputations due to (a) the platform's flexibility, which allows instant one-day bookings to occur on the same day, and (b) the challenge of obtaining a ground truth to validate the imputation process.

>>Present a clear analysis of the potential impact of data quality issues on the results. This could involve sensitivity analyses that assess how variations in data quality might affect key findings.

Authors response: Thanks for this suggestion. We agree that it is essential to address the data quality issue when presenting our proposed method and applying it in the case study. Despite a lack of ground truth and raw scraping datasets from other scraping exercises, we respond to the variation of data quality issues in Page 20. The less frequent data releases, hence with lower data quality, are simulated by sampling on our scraped data. Table 4 demonstrates the detailed summary of such simulations comparing our results with weekly and fortnightly samplings from our data. Our analysis shows that data with various qualities do have an impact on the estimation results. Weekly sampling shows fluctuations while fortnightly sampling has substantially lower estimations. We added ‘The findings highlight the potential influence that varying data quality could have on such analyses’ in line 497 (Page 20) to signpost this important aspect addressed by this section.

-Platform and Contextual Changes: The study focuses on a specific time period (COVID-19 pandemic) and a specific location (Edinburgh). The discussion should include considerations about the generalizability of findings to other time periods, locations, and different regulatory contexts.

Please discuss about:

>>Address the potential influence of platform changes (such as modifications to Airbnb's calendar format) on the data collection and analysis process. Explain how the proposed methodology adapts to such changes and maintains its accuracy.

Authors response: We agree that this aspect is important for ensuring the long-term sustainability of our method. Our scraping process acquires access to publicly available consumer-facing web content and retrieves daily updates from the booking calendar. This procedure relies on interfacing with the current Airbnb API, responsible for delivering web content to consumers. Our data collection efforts have spanned from 2020 until the present, and the likelihood of significant alterations to the data structure of this API is quite low. Nevertheless, our dedication to continuous close monitoring of the website's responses remains, allowing us to promptly adapt our scraping process as needed. Methodologically, the method proposed in this paper represents a mathematical abstraction of booking updates, making it adaptable to potential future changes.

We added the ‘Technically, our scraping process accesses public consumer-facing web content daily through the current Airbnb API since 2020. A vigilant monitoring of website responses to the scraping approach is needed, and our method and abstract data structure are designed to easily accommodate any future changes’ to line 545 (Page 22). 

>>Discuss the transferability of the approach to different locations and time periods, along with potential challenges and adjustments required to apply the methodology in diverse contexts.

Authors response: Thanks for this suggestion. This is a great point adding to our contribution. While it demands initial resources for setup, we have made the scraping exercise codebase publicly available at [https://github.com/urbanbigdatacentre/ubdc-airbnb/tree/master/README]. The scraping exercises have been extended to cover the whole Scotland and 8 cities across the UK since 2020. It is readily extensible to other cities worldwide.

The data method used to extract the calendar updates and related indicators is also available at the author’s github [https://github.com/YangWang-Glasgow/Airbnb-Processing-Daily-Booking-Calendars]. The extracted indicators can be used for diverse research and regulation purposes.

We have added ‘The web scraping code (https://github.com/urbanbigdatacentre/ubdc-airbnb/tree/master/README) and data methods (https://github.com/YangWang-Glasgow/Airbnb-Processing-Daily-Booking-Calendars) have been openly available. They are easily extensible to other cities worldwide, effectively addressing both regulatory and research requirements’ in line 541 (Page 22). 

Conclusion:

The paper presents a valuable approach to address the limitations of existing data sources in studying short-term rental activity, particularly in the context of Airbnb. The methodology's potential benefits for policy evaluation and market analysis are clear. However, to strengthen the paper, it is essential to address the weaknesses mentioned above. More in-depth discussions of the methodology's limitations, data quality issues, and the broader applicability of findings would provide a more comprehensive understanding of the approach's strengths and limitations. Additionally, further explanation of assumptions, potential biases, and strategies to handle missing data would enhance the paper's credibility and usefulness to both researchers and policymakers.

Authors response: We have made the corresponding updates to the paper to address these recommended issues.

---

## [Decision Letter · Decision Letter 1]

21 Jan 2024

Enhancing our Understanding of Short-term Rental Activity: A Daily Scrape-based Approach for Airbnb Listings

PONE-D-23-19176R1

Dear Dr. Wang,

We’re pleased to inform you that your manuscript has been judged scientifically suitable for publication and will be formally accepted for publication once it meets all outstanding technical requirements.

Kind regards,

Sutee Anantsuksomsri

Academic Editor

PLOS ONE

Additional Editor Comments (optional):

The authors responded to all comments and suggestions from reviewers and the academic editor. I recommend accepting the manuscript for publication.

Reviewers' comments:

Reviewer's Responses to Questions

**Comments to the Author**

1. If the authors have adequately addressed your comments raised in a previous round of review and you feel that this manuscript is now acceptable for publication, you may indicate that here to bypass the “Comments to the Author” section, enter your conflict of interest statement in the “Confidential to Editor” section, and submit your "Accept" recommendation.

Reviewer #1: All comments have been addressed

2. Is the manuscript technically sound, and do the data support the conclusions?

Reviewer #1: Yes

3. Has the statistical analysis been performed appropriately and rigorously? 

Reviewer #1: Yes

4. Have the authors made all data underlying the findings in their manuscript fully available?

Reviewer #1: Yes

5. Is the manuscript presented in an intelligible fashion and written in standard English?

Reviewer #1: Yes

6. Review Comments to the Author

Reviewer #1: (No Response)

7. PLOS authors have the option to publish the peer review history of their article (what does this mean?). If published, this will include your full peer review and any attached files.

Reviewer #1: No

---

## [Editor Report · Acceptance letter]

30 Jan 2024

PONE-D-23-19176R1 

PLOS ONE

Dear Dr. Wang, 

I'm pleased to inform you that your manuscript has been deemed suitable for publication in PLOS ONE. Congratulations! Your manuscript is now being handed over to our production team.

Kind regards, 

on behalf of

Dr. Sutee Anantsuksomsri 

Academic Editor

PLOS ONE